# Differential Effects of Short- and Long-Term Negative Affect on Smartphone Usage: The Moderating Role of Locus of Control

**DOI:** 10.3390/bs15081121

**Published:** 2025-08-18

**Authors:** Yang Chu, Jiahao Li, Shan Liu, Yanfang Liu, Jie Xu

**Affiliations:** 1Center for Psychological Sciences, Zhejiang University, Hangzhou 310058, China; yangchu@zju.edu.cn (Y.C.); jiahao_li@zju.edu.cn (J.L.); 13235966023@163.com (S.L.); 2Huawei Technologies Co., Ltd., Beijing 100080, China; liuyanfang2@huawei.com; 3Department of Psychology, Lingnan University, Hong Kong SAR, China

**Keywords:** smartphone use, negative affect, psychological distress, daily mood, locus of control, emotional regulation, daily diary method

## Abstract

In the digital age, smartphones are often used as tools for emotion regulation. While prior research has examined affective predictors of smartphone use, few studies have considered the combined impact of short-term and long-term affective states. This study investigates how daily negative emotional states and psychological distress relate to smartphone use and whether these associations are moderated by locus of control, a core belief about perceived control. Thirty-seven participants completed a one-month daily diary study combined with objective smartphone usage tracking, which yielded 837 valid observations. Multilevel analyses showed no association between daily negative emotional state and smartphone use. However, psychological distress predicted divergent behavioral patterns based on locus of control: individuals with an internal locus of control showed reduced usage under distress, whereas those with an external locus of control exhibited increased frequency of use. These findings highlight the importance of individual control beliefs in shaping technology-mediated emotion regulation and offer implications for interventions targeting excessive smartphone use.

## 1. Introduction

Smartphones have become essential in modern work and daily life. Despite their benefits, smartphones are also linked to a range of negative consequences, which have raised increasing concerns. These negative outcomes, including reduced attention, poor sleep quality, and mental health issues ([84]), may escalate into problematic smartphone use or even smartphone addiction ([10]; [31]). These concerns have driven researchers to investigate not only the consequences but also the underlying determinants of smartphone use, particularly in relation to its role in regulating emotions. Often referred to as a “security blanket” or “first aid in the pocket”, smartphones help individuals cope with daily negative affect ([53]; [66]). Although studies have pointed to a link between negative affect (e.g., depression, anxiety) and excessive smartphone use ([11]; [71]), uncertainties remain regarding the precise nature of this relationship. Most existing work has examined either short-term or long-term negative affect in isolation ([14]; [9]; [63]), with little research clarifying their distinct effects. Furthermore, many studies have relied upon self-reported data and concentrated on relatively brief smartphone use periods ([31]; [15]; [83]), thereby restricting their ability to capture comprehensive patterns of actual smartphone use over time.

In order to address these gaps, we conducted a one-month objective tracking of smartphone usage, incorporating daily negative emotional state assessments and a single-time measure of psychological distress. Additionally, we examined individual differences in locus of control (LOC) to investigate its moderating role in the relationship between negative affect and smartphone use. By integrating these elements, the present study advances the understanding of the impact of negative affect on smartphone usage and provides insights for effective interventions to mitigate excessive use.

### 1.1. Relationship Between Negative Affect and Smartphone Use

Uses and Gratifications Theory (UGT) provides a foundational framework for understanding why individuals use media ([35]). UGT posits that users are active individuals who selectively use media to obtain specific “gratifications” that fulfill their needs. One key gratification among these is stress relief and the regulation of emotions and mood ([81]). Compensatory Internet Use Theory (CIUT) extends UGT ([80]) by proposing that when individuals face negative situations in real life, including stress, frustration, or unmet needs, they may utilize the Internet as a compensatory mechanism to alleviate negative affect ([34]). CIUT offers an essential framework for understanding the relationship between negative affect and smartphone use.

Negative affect is a broad term that encompasses a variety of negative feelings. Based on temporal stability, it ranges from state-like to trait-like forms ([22]). In this context, CIUT explains the relationship between negative affect and smartphone use by addressing both the short-term emotional states that influence an individual’s smartphone use (within-person processes) and the long-term negative affect that contributes to individual distinctions in smartphone use (between-person processes). Thus far, most studies supporting CIUT have employed cross-sectional designs ([14]; [9]), concentrating on the perspective of individual differences by measuring psychological distress (such as depression, anxiety, and stress). Psychological distress is considered a stable, long-term form of negative affect ([61]) and is related to increased smartphone use ([10]; [33]). However, some studies have also focused on the direct impact of individuals’ short-term emotional fluctuations on smartphone use ([63]; [83]; [85]). For instance, researchers found an association between daily moods and smartphone use ([63]; [83]).

In summary, while existing literature has separately confirmed that both long-term (trait-like) and short-term (state-like) negative affect are associated with smartphone use, few studies have situated both within a unified framework to systematically investigate the potentially distinct psychological mechanisms that may underlie them. Clarifying these mechanisms is crucial for developing effective interventions ([14]).

### 1.2. Distinguishing Compensatory Use Mechanisms with a Dual-System Framework

Building on the gap identified above, this study proposes two distinct compensatory mechanisms. The first is an emotion-driven mechanism, where smartphone use is directly driven by short-term emotions, and it represents a within-person (time-varying) process. The second is habit-driven behavior, where long-term negative affect reinforces smartphone use ([10]), which becomes less sensitive to momentary emotional fluctuations. It reflects a between-person (time-invariant) process that focuses on the automation and stability of behavior.

The dual-system model provides a useful lens for understanding these differences. It distinguishes two distinct psychological processing systems: a fast, automatic, and emotion-driven system (the automatic system), which governs immediate emotional reactions and rapid reward valuation; and a slow, controlled, and goal-oriented system (the deliberative system), responsible for impulse inhibition, self-regulation, and planned behavior ([20]; [32]; [47]; [67]). This model is widely applied to impulse control and behavioral addiction and is supported by neural substrates ([67]). Specifically, the emotion-driven mechanism involves the dynamic interplay of both systems ([28]). The automatic system rapidly generates instinctive emotional impulses, while the deliberative system may inhibit these impulses or strategically guide smartphone use toward emotion-regulation goals ([26]). The final usage behavior is, thus, an integrated outcome of impulse and regulation. By contrast, the habit-driven mechanism is primarily governed by the automatic system. When individuals experience persistent negative affect, the repeated use of a smartphone to gain temporary relief constitutes negative reinforcement ([3]). Over time, such use shifts from goal-directed action to a stimulus–response habit, becoming largely independent of deliberative oversight ([82]).

The dual-system model not only explains the divergence between emotion-driven and habit-driven use but also emphasizes the central role of self-regulation in determining which pathway predominates. Self-regulation capacity is not random; it is systematically shaped by stable personality traits. Among these traits, LOC offers a theoretically grounded way to account for individual differences in how the two systems are engaged ([1]).

### 1.3. The Moderating Role of Locus of Control

Locus of control (LOC), originally proposed by [64] ([64]), refers to an individual’s belief regarding the extent to which their environment is controllable and outcomes are responsive to their actions. LOC is a unidimensional construct that ranges from internal to external ([17]). Individuals with an internal LOC feel that they can influence outcomes through their own actions, while those with an external LOC typically believe that outcomes are more determined by external circumstances ([7]). LOC is a critical personality trait which has been shown to reliably predict affect, cognition, and behavior ([21]). It significantly shapes how individuals manage stress and influences behavioral patterns ([17]; [56]).

Individuals with an internal LOC commonly engage in problem-focused coping and actively address stressors to mitigate their impact ([56]; [77]). Conversely, individuals with an external LOC are more likely to rely upon emotion-focused coping, employing suppression strategies or external methods to handle emotions and stress ([50]). Furthermore, LOC influences how individuals engage with technology ([19]; [55]). Individuals with an internal LOC may utilize smartphones more strategically, employing them as tools for obtaining information and exerting control over their environment ([6]). Conversely, those with an external LOC are more prone to problematic internet or smartphone use ([30]; [39]). For instance, [40] ([40]) revealed that while LOC does not influence the overall amount of smartphone use, it is significantly related to inappropriate use at unsuitable times. Similarly, [8] ([8]) reported a positive correlation between external LOC and nomophobia.

Expanding on these findings, this study hypothesizes that LOC moderates the relationship between negative affect and smartphone use. Individuals with an internal LOC are more likely to engage their deliberative system, consciously choosing more adaptive coping strategies and thereby decreasing their dependence on smartphones. Conversely, those with an external LOC may be more susceptible to the automatic system, which leads them to increase their smartphone use as a maladaptive coping mechanism for managing negative affect.

### 1.4. Smartphone Use Indicators

Smartphone use can be quantified by duration (the total time spent using smartphones) and frequency (the number of usage episodes) ([29]; [74]). Although both reflect usage intensity, they may be driven by different psychological factors. [74] ([74]) argued that duration often reflects an intentional investment of time rooted in intrinsic motivation, whereas frequency is more determined by external factors and environmental demands. Empirical evidence showed that their shared variance is often limited ([71]), indicating that they capture partially distinct behavioral patterns ([72]).

Frequency is often regarded as a direct proxy for measuring the strength of habitual behavior ([49]; [75]). In the context of smartphone use, one of the most common habitual patterns is the checking habit ([62]), which typically involves constant checking or participation in a cycle of smartphone checking ([86]). Checking behavior is defined as brief and frequent inspections of dynamic content ([52]). Such high-frequency, short-duration usage patterns reflect the impulsive and automatic nature of habitual use ([23]). Moreover, frequency and duration of use are functionally interrelated. For example, [79] ([79]) reported that fragmented checking—even when each session is brief—can accumulate to substantially increase overall screen time. Measuring both indicators enables the analysis of average session duration (i.e., the amount of time spent in a single usage session; [85]). Notably, very short average session durations typically suggest unconscious checking, whereas longer average sessions are more likely to reflect deliberate engagement.

To accurately capture these complementary behavioral patterns and circumvent the systemic biases of self-reports ([46]), the present study incorporated objective tracking of both daily smartphone use duration and frequency. This dual-indicator approach enables us to test whether the proposed effects of negative affect and LOC are robust across distinct facets of smartphone use and to explore potential divergences between duration-based and frequency-based outcomes.

### 1.5. The Present Study

Building on this review of the prior literature, this study seeks to examine the determinants of smartphone use, focusing on short-term negative affect (daily negative emotional state, DNES), long-term negative affect (psychological distress, PD), and the moderating role of LOC. The present study employs a one-month daily diary design with continuous smartphone use tracking. We propose the following hypotheses:

**H1** **(Within-person effect).**
*On days when individuals report a higher DNES, their smartphone use will be positively correlated.*


**H2** **(Between-person effect).**
*Individuals with higher levels of PD will exhibit higher smartphone use.*


**H3** **(Cross-level moderation effect).**
*LOC will moderate the within-person association between DNES and smartphone use. Specifically, this positive association will be stronger for individuals with a higher external LOC.*


**H4** **(Between-person moderation effect).**
*LOC will moderate the between-person association between PD and smartphone use. Specifically, this positive association will be stronger for individuals with a higher external LOC.*


Additionally, we explore whether the observed patterns differ depending on whether smartphone usage is measured by duration or frequency.

## 2. Methods and Materials

### 2.1. Participants and Procedure

This study recruited 40 participants from eastern China through online convenience sampling. It was conducted from November 2021 to January 2022 using a rolling recruitment strategy, ensuring that each participant took part for approximately one month. Three participants withdrew during the study, resulting in a final sample of 37 participants. Participants were aged between 18 and 39 (M = 26, SD = 5.02), with 48.6% being female. The study protocol was approved by the Institutional Review Board.

The study consisted of two main components: (1) a daily diary, in which participants completed a DNES report each day; and (2) continuous smartphone use tracking, in which a custom Android application automatically recorded the app name, start time, and duration of each user-initiated smartphone use session. At the start of the study, participants also completed questionnaires to gather person-level information (e.g., LOC).

Participants had an average of 25.1 participation days (SD = 7.0), which yielded a total of 929 study days. The rate of valid day-level observations (i.e., days with both smartphone tracking and diary entries) was 90.1% (837 out of 929). Thus, participants contributed an average of 22.6 complete day-level observations. Previous research indicates that multiple imputation of missing values does not offer benefits for mixed-effects modeling ([18]). Therefore, listwise deletion was ultimately chosen for the main analyses in this study.

### 2.2. Measures

DNES was assessed each evening at 10 p.m. to capture participants’ reflections on their overall emotional experiences of the day, consistent with prior daily diary methods ([83]). Participants completed a single-item online questionnaire, rating their daily emotional state on a 5-point scale ranging from 1 (“Very Negative”) to 5 (“Very Positive”). For analysis, these scores were reverse-coded so that higher values indicated greater negative affect, consistent with the study’s focus on negative emotional states. This brief single-item measure, adapted from previous studies (e.g., [85]), was selected to reduce response burden.

PD was measured using the Depression Anxiety Stress Scale (DASS-21) ([51]). The scale consists of three subscales—depression, anxiety, and stress—each comprising seven items. Example items include the following: “I couldn’t seem to experience any positive feeling at all” (depression), “I experienced breathing difficulty” (anxiety), and “I found it hard to wind down” (stress). Items are rated on a 4-point scale, which ranges from 0 (“Did not apply to me at all”) to 3 (“Applied to me very much, or most of the time”), with higher scores indicating greater symptom frequency. Following the DASS-21 manual ([73]), raw scores from the three subscales were standardized into Z-scores to derive an overall PD score. The Cronbach’s alpha for the sample was 0.92.

LOC was measured using the Internal-External Locus of Control Scale ([64]), which includes 29 forced-choice items, with 6 unscored filler items. For each item, two options are presented, representing internal or external control orientations. Scores are calculated by totaling the items indicative of external control, resulting in an overall score ranging from 0 to 23. Higher scores indicate external LOC, while lower scores reflect internal LOC ([70]). A representative item is “Becoming a success is a matter of hard work, luck has little or nothing to do with it” (internal) versus “Getting a good job depends mainly on being in the right place at the right time” (external). The Cronbach’s alpha for this study was 0.72.

Demographics, such as gender, age, and occupation, were collected. Additionally, other measures—including those related to personality—were administered but excluded from the present analysis, as they were beyond the scope of the study.

Smartphone use was assessed as follows: (1) daily smartphone use duration (DSUD) refers to the total amount of time (in minutes) that participants used smartphone applications each day, and (2) daily smartphone use frequency (DSUF) represents the number of times participants launched any applications daily.

### 2.3. Data Analysis

To account for the nested structure of repeated longitudinal data, a multilevel modeling framework was employed ([68]). Two separate models were constructed for each DSUD and DSUF, using the same predictors. Model 1 tested the main effects, while Model 2 examined the moderation hypotheses. The model equations are as follows:

Model 1 (Main Effects):(1)$$\left\{{DSUD}_{it},{DSUF}_{it}\right\}=\beta_{0i}+\beta_{1i}\;\times \;wp.DNES_{it}+\varepsilon_{it}$$
(2)$$\beta_{0i}=\gamma_{00}+\gamma_{01}\times bp.DNES_{i}+\gamma_{02}\times PD_{i}+\gamma_{03}\times LOC_{i}+\gamma_{04}\times Age_{i}\;+\gamma_{05}\times Gender_{i}+u_{0i}$$(3)$$\beta_{1i}=\gamma_{10}$$

Model 2 (Interaction Effects):(4)$$\left\{{DSUD}_{it},{DSUF}_{it}\right\}=\beta_{0i}+\beta_{1i}\times wp.DNES_{it}+\varepsilon_{it}$$(5)$$\beta_{0i}=\gamma_{00}+\gamma_{01}\times bp.DNES_{i}+\gamma_{02}\times PD_{i}+\gamma_{03}\times LOC_{i}+\gamma_{04}\times Age_{i}\;+\gamma_{05}\times Gender_{i}+\gamma_{07}PD_{i}\times LOC_{i}+u_{0i}$$(6)$$\beta_{1i}=\gamma_{10}+\gamma_{11}LOC_{i}$$

${DSUD}_{it}\;and\;{DSUF}_{it}$ represent the repeated measures of smartphone use for individual *i* on day *t*. The intercept ($\beta_{0i}$) reflects each individual’s baseline level of smartphone use, while the slope ($\beta_{1i}$) captures the within-person effect of a time-varying predictor. The fixed-effect coefficients (γ) estimate average associations at the sample level, and the random effects (u) are the residual unexplained between-person differences. The DNES predictor was decomposed into within-person fluctuations (wp.DNES) and between-person differences (bp.DNES). Model 1 included wp.DNES and PD as predictors, focusing on their main effects on smartphone use. Model 2 expanded Model 1 by incorporating the interaction terms (wp.DNES × LOC and PD × LOC) to examine LOC’s moderating effects.

Data analysis was conducted using R (Version 4.2.2), particularly the lme4 statistical package. All Level 1 predictors were group-mean centered, and all Level 2 predictors were grand-mean centered ([16]). Furthermore, based on prior research indicating that age and gender are important demographic factors in smartphone use ([48]; [54]), these two variables were included as control variables in our analyses.

## 3. Results

### 3.1. Preliminary Analysis

Table 1 presents the descriptive statistics and correlations among the variables. On average, participants used their smartphones for 308.15 min per day (DSUD) and launched apps 184 times per day (DSUF). The two smartphone usage metrics were strongly correlated at both the between-person (r = 0.81, *p* < 0.001) and within-person levels (r = 0.77, *p* < 0.001). As for the measures of negative affect, the average monthly DNES and PD were positively associated (r = 0.33, *p* = 0.049).

Regarding the core hypotheses, there were no significant correlations between PD and the average smartphone usage metrics (Avg DSUD and Avg DSUF) at the between-person level. Similarly, at the within-person level, daily DNES was not significantly correlated with daily smartphone usage. Among the control variables, age was positively correlated with average DSUD (r = 0.39, *p* = 0.019) but not with DSUF. Age was also negatively correlated with average DNES (r = −0.33, *p* = 0.045). Gender was not significantly associated with any of the main study variables.

Finally, variance partitioning revealed intraclass correlation coefficients of 0.31 for DNES, 0.64 for DSUD, and 0.64 for DSUF, indicating substantial between-person variability and supporting the use of multilevel modeling.

### 3.2. Multilevel Models

The results of the multilevel linear models are presented in Table 2 and Table 3.

Model 1: In terms of DSUD, the effect of wp.DNES (β = 0.03, *p* = 0.151) and PD (β = 0.07, *p* = 0.603) was not significant. For DSUF, neither wp.DNES (β = −0.01, *p* = 0.829) nor PD (β = −0.05, *p* = 0.840) showed significant effects. The control variables of age and gender also did not show any significant effects on either DSUD or DSUF (all *p*s > 0.05).

Model 2: The interaction effect of wp.DNES × LOC was not significant for DSUD (β = −0.02, *p* = 0.312) or DSUF (β = −0.03, *p* = 0.194). The interaction effect of PD × LOC was marginally significant for DSUD (β = 0.27, *p* = 0.074) but was significant for DSUF (β = 0.47, *p* < 0.001). Age had a significant positive effect on DSUD (β = 0.34, *p* = 0.020) but not on DSUF. Gender had no significant effect on either usage metric.

To further probe the significant interaction, we conducted a simple slopes analysis. Figure 1 presents the effect of PD on smartphone use for low LOC (−1 SD, internal LOC) and high LOC (+1 SD, external LOC) ([70]). Visually, as LOC becomes more external, the effect of PD on smartphone use shifts from negative to positive. These patterns were significant for DSUF and marginal for DSUD. Simple slopes analyses for DSUF revealed that for individuals with an internal LOC, PD had a significantly negative effect on DSUF (β = −15.26, *p* = 0.012), while with an external LOC, PD had a significantly positive effect on DSUF (β= 16.81, *p* = 0.014).

Additionally, we examined the daily average session duration (calculated as DSUD divided by DSUF) as a complementary indicator of smartphone use. The interaction effect of PD × LOC was significant for average session duration (β = −0.24, *p* = 0.03; Figure 2), suggesting that under high PD, individuals with an internal LOC tended to have longer smartphone use per session compared to those with an external LOC—a pattern that was not significant under low PD.

## 4. Discussion

In order to investigate the effects of negative affect and LOC on smartphone use, this study employed a longitudinal design and objective data tracking. The results demonstrated no significant within-person association between DNES and smartphone use. However, LOC significantly moderated the relationship between PD and daily smartphone use frequency and showed a marginally significant moderating effect on daily smartphone use duration. These results offer a nuanced perspective on the complex relationships between negative affect, personality traits, and smartphone use patterns. This serves to contribute to a deeper understanding of CIUT.

### 4.1. The Limited Efficacy of the Emotion-Driven Pathway

This study found no significant within-person association between DNES and smartphone use (H1 was not supported). These results align with prior research that failed to detect significant within-person associations between short-term negative affect and smartphone use ([2]; [85]). As [2] ([2]) suggested, the relationship between emotional fluctuations and smartphone use is unlikely to follow a simple “dose-response” pattern. This finding can be interpreted through the lens of the dual-system model, which highlights the significant role of the deliberative system. The deliberative system manages emotional impulses to produce divergent, goal-directed outcomes ([67]). For example, in response to negative emotions, an individual might strategically use their smartphone for social support (promoting use), while at another moment, they might consciously inhibit the same urge to focus on a demanding task. It is plausible that these opposing, consciously driven behaviors co-exist throughout the day, thereby canceling each other out in aggregated daily-level analyses. This mechanism also helps to explain the importance of timescale. For example, [63] ([63]) observed a positive association between short-term emotional states and usage when participants rated their emotions 10 times during the day. This suggests that momentary emotional urges may not systematically translate into an overall increase in daily smartphone use.

Building upon this, this study did not find a significant cross-level moderating effect of LOC on the relationship between DNES and smartphone use (H3 was not supported). The absence of a significant main effect (H1) means there was no consistent, systematic association for a moderator to act upon. Additionally, the stable dispositional trait LOC may have a limited role in moderating immediate, state-level reactions to daily emotional variations ([21]). It is plausible that reactions to daily emotions are more likely influenced by situational constraints (e.g., workload, social obligations; [25]). Instead, LOC is better suited for predicting responses to chronic, persistent stressors, such as PD.

Therefore, the absence of a direct within-person effect at the daily level does not necessarily invalidate the emotion-driven pathway proposed by CIUT. Rather, it highlights its nature as a momentary, highly context-dependent process that may not aggregate into a detectable increase in total daily use.

### 4.2. The Habit-Driven Pathway: Locus of Control Shapes Compensatory Use

This study found no between-person association between PD and smartphone use (H2 was not supported). This finding aligns with the notable lack of consensus in the literature, which has reported both positive ([10]; [33]) and negative ([12]) correlations between PD and smartphone use. We argue that this inconsistency reflects the complexity of the relationship rather than its absence, specifically, conditional effects shaped by personality. Furthermore, the methodological differences in the literature likely contributed to these divergent results. While the present study employed objective smartphone usage data, most prior studies have relied on self-report measures. Prior research comparing self-reported and objectively logged digital behaviors has demonstrated that self-reports often misestimate actual usage ([15]), potentially obscuring moderation effects that may be detectable with more accurate measures.

This study identified a significant interaction between PD and LOC on smartphone use frequency (providing support for H4). Although this interaction was not significant for duration, a similar pattern emerged. Specifically, individuals with an external LOC increased their smartphone use under high PD levels, whilst those with an internal LOC typically reduced their usage. This result is consistent with prior research indicating that coping strategies are influenced by LOC ([17]; [58]). Moreover, this finding supports the notion that individuals with an external LOC are more prone to maladaptive usage of digital devices ([30]; [39]). To understand this interaction pattern, it is essential to consider the coping tendencies typically associated with LOC. External LOC involves attributing problems to uncontrollable external factors ([36]), lowering perceived personal control and fostering a problem-avoidance orientation ([64]). Such orientation is associated with escape–avoidance coping ([37]) and self-distraction ([5]), which aim to distract negative emotions rather than attempt problem resolution ([69]). Smartphones, with their mobility, connectivity, and multifunctionality ([76]), are well suited to support emotion regulation strategies, such as mood relief ([65]), digital procrastination ([60]), and social reassurance seeking ([13]; [59]). Over time, the repeated use of smartphones for emotional escape can reinforce these behaviors, gradually solidifying them into automatic habits ([10]). From a dual-system perspective, habit formation reduces the reliance on deliberative processes, thereby increasing the influence of impulsive, automatic processes. This habit-driven mechanism offers a plausible explanation for the significantly elevated smartphone use observed among individuals with an external LOC under high psychological distress. Conversely, individuals with an internal LOC—who believe they can control their environment—typically adopt active, problem-focused coping strategies ([50]; [77]). In high-distress conditions, these individuals may deliberately limit smartphone use to avoid distraction and preserve cognitive resources for problem solving ([38]). Such an approach requires deliberate planning and sustained attention; processes governed by deliberate systems to regulate impulses and serve long-term goals.

Taken together, our findings suggest that long-term negative affect—reflected in between-person differences—plays a more substantial role in driving overall daily usage than short-term emotional fluctuations. This pattern suggests the habit-driven mechanisms within CIUT are more relevant to smartphone use behaviors than the emotion-driven mechanisms.

### 4.3. The Meaning of Different Smartphone Use Indicators

DSUD and DSUF measure how much versus how often an individual uses their smartphone in a single day. The average levels of these indicators were consistent with those reported in previous studies (e.g., [2]; [27]), providing evidence for the validity of our measurements and the generalizability of our findings. Although DSUD and DSUF were correlated, their divergent statistical patterns in this study suggest that they capture partially overlapping yet distinct behavioral tendencies.

Frequency of behavior is a key indicator of habit strength ([75]). Consistent with this, our findings show that smartphone use frequency is more strongly influenced by the interaction between PD and LOC than duration. This suggests that frequency of use may be a more direct expression of habit-driven patterns than total duration. Moreover, under high PD, individuals with an external LOC exhibited significantly shorter daily average session durations than those with an internal LOC, which may reflect checking behaviors. This pattern supports research suggesting that checking behaviors are habitual responses to reduce the gap between an individual’s perceived and actual state in the real world ([23]). For individuals with an external LOC, such checking behaviors may serve as a coping mechanism to manage distress and reduce uncertainty ([42]).

Demographic variables further help distinguish DSUD and DSUF. In our study, age was a significant positive predictor of smartphone use duration but not frequency. This pattern suggests that older users may engage in longer, more sustained sessions of phone use—such as news reading ([4])—rather than the high-frequency, brief checking behaviors typically reflected in DSUF. Consistent with this interpretation, prior objective logging research has shown that older adults have significantly longer average app session durations compared to younger users ([24]).

Taken together, these results reinforce our central argument that duration and frequency are not interchangeable metrics. They likely reflect different underlying motivations and user profiles, with frequency serving as a more direct behavioral expression of the habit-driven processes examined in this study.

### 4.4. Implications

CIUT posits that individuals use smartphones to alleviate negative affect. Building on this, our study specifies and refines CIUT by situating its dual pathways within a dual-system framework. We found no evidence that daily emotional fluctuations systematically drive overall usage, likely because immediate affective impulses are variably regulated by reflective processes. By contrast, our results highlight the relevance of the habit-driven mechanism—a process reflecting the dominance of automatic impulses—which is shaped by the interplay between persistent psychological distress and an individual’s locus of control. This suggests that, within CIUT, long-term emotional vulnerabilities and stable individual differences are more influential in driving compensatory smartphone use than transient emotional states.

Given these differentiated pathways, interventions to mitigate excessive smartphone use should distinguish between short- and long-term negative affect. In terms of short-term negative affect, certain interventions, such as personalized notifications, can help promote immediate self-regulation ([44]). Conversely, more comprehensive support is required to address long-term negative affect, including coping-skills training ([41]) and programs focused on bolstering individuals’ LOC ([87]). By addressing various affective factors, interventions can be more tailored and effective.

Moreover, this study emphasizes the importance of measurements of smartphone use. In comparison to self-reported data, logged data provides a more precise reflection of actual behavior ([57]). Additionally, a multifaceted measurement approach—disaggregating smartphone use into components (e.g., frequency, duration, session length)—could help capture the psychological mechanisms.

### 4.5. Limitations and Future Study

Despite its strengths, this study had several limitations. First, a significant limitation is the use of a single-item measure for DNES. While intended to reduce participant burden, this approach compromises construct validity, as negative affect is a multifaceted construct that a single item cannot fully capture. Crucially, this may limit the interpretability of our null findings for within-person effects. The measure might lack the sensitivity to detect subtle daily emotional fluctuations. Therefore, the absence of a significant effect could be an artifact of this measurement weakness. Future research should employ validated short-form scales (e.g., the short-form PANAS; [45]) to allow for a more conclusive test of these hypotheses. Second, despite employing a longitudinal design, this study only measured participants’ emotional state at the end of each day. Consequently, it was not possible to draw conclusions about whether affect influences smartphone use or vice versa ([10]). Future studies might consider more intensive measurements or advanced analytical methods to better capture temporal dynamics and reinforce causal inferences. Third, the sample size was relatively small and consisted primarily of Android users, which may have restricted the external validity of the findings. Future research should diversify the sample to enhance the generalizability and reliability of the results. Finally, although this study utilized objective smartphone use measures, it did not capture the broader contexts in which smartphones are used ([78]). Going forward, studies should refine behavioral indicators by considering factors such as usage categories and contexts ([43]). This could help to identify more precise patterns of smartphone usage and offer deeper insights into the interactive mechanisms between affect and smartphone use.

## 5. Conclusions

Based on CIUT, this study used objective measurements and a longitudinal design to examine the impact of short- and long-term negative affect on smartphone use. Additionally, this study incorporated LOC as a moderating factor. The results showed (1) no significant association between DNES and daily smartphone use and (2) that LOC moderated the relationship between PD and smartphone use frequency. These findings underscore the crucial role of habituation mechanisms rather than emotional triggers in smartphone use. By integrating both short- and long-term emotional factors alongside individual differences, this study provides valuable insights for developing more targeted interventions in problematic smartphone use.

## Figures and Tables

**Figure 1 behavsci-15-01121-f001:**
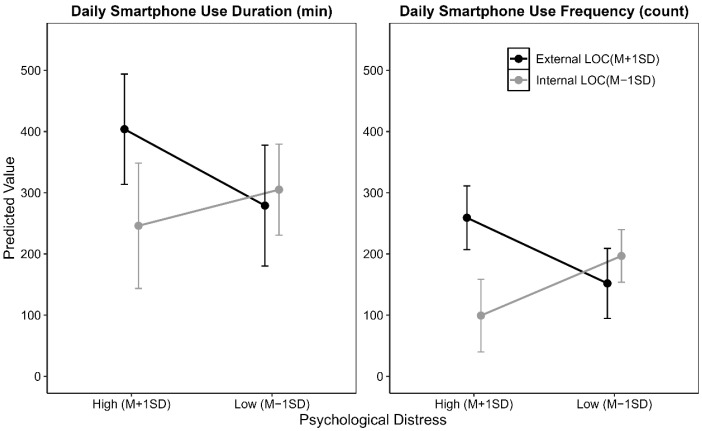
The moderation effect of locus of control in the relationship between psychological distress and daily smartphone use. LOC = locus of control.

**Figure 2 behavsci-15-01121-f002:**
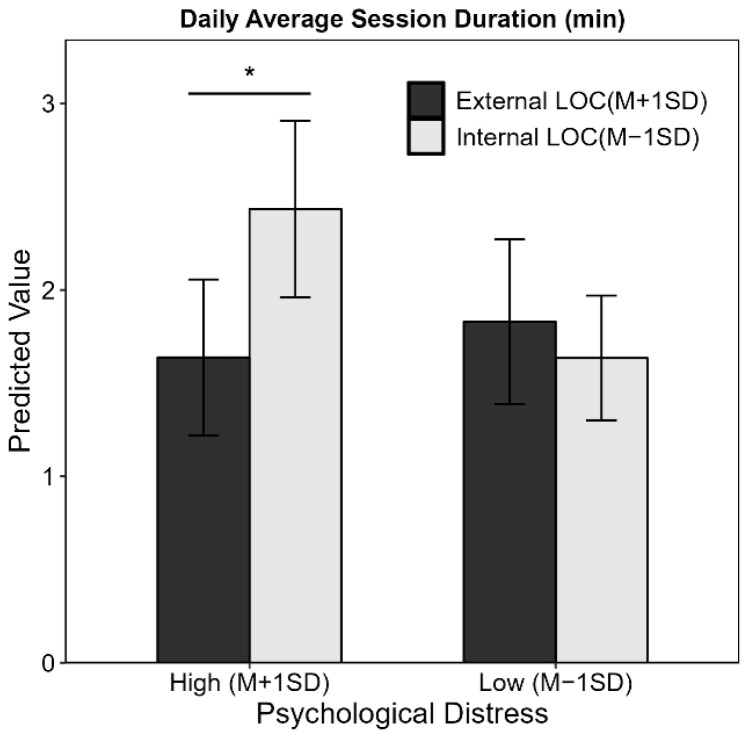
The moderation effect of locus of control in the relationship between psychological distress and daily average session duration. The daily average session duration is calculated by dividing the total smartphone use time by the number of smartphone use sessions. LOC = locus of control. * *p* < 0.05.

**Table 1 behavsci-15-01121-t001:** Descriptive statistics for the primary measures at the between-person and within-person levels.

Variable	M	SD	1	2	3	4	5	6
Between-person								
1. Age	26.00	5.02						
2. Gender	−0.01	0.51	−0.15					
3. PD	4.65	3.42	−0.13	0.16				
4. LOC	10.73	3.97	0.12	−0.02	0.18			
5. Avg DNES	2.64	0.53	−0.33 *	0.23	0.33 *	0.24		
6. Avg DSUD	308.15	139.71	0.39 *	0.01	0.03	0.21	−0.19	
7. Avg DSUF	184.02	87.73	0.16	0.18	−0.05	0.19	−0.18	0.81 **
Within-person								
1. DNES								
2. DSUD			0.05					
3. DSUF			0.02	0.77 **				

Observations = 837, N = 37. Gender: 1/2 = female, −1/2 = male; PD (psychological distress) is the average of the depression, anxiety, and stress subscales; Avg indicates each participant’s average level of the variable across the study period; LOC = locus of control; DNES = daily negative emotional state; DSUD = daily smartphone use duration; DSUF = daily smartphone use frequency. * *p* < 0.05. ** *p* < 0.01.

**Table 2 behavsci-15-01121-t002:** Multilevel linear model (Model 1) results for daily smartphone use duration and frequency.

	DSUD			DSUF		
Predictors	*β*	*t*	*p*	*β*	*t*	*p*
wp.DNES	0.03	1.44	0.151	0.00	0.22	0.829
bp.DNES	−0.13	−0.91	0.365	−0.20	−1.35	0.178
PD	0.07	0.52	0.603	−0.03	−0.20	0.840
LOC	0.16	1.12	0.263	0.20	1.41	0.158
Age	0.29	1.93	0.054	0.07	0.45	0.650
Gender	0.07	0.53	0.593	0.21	1.52	0.129
	Random Effects					
σ^2^	110,678.80			4194.87		
τ_00_	17,398.95			7355.40		
ICC	0.62			0.64		
N	37			37		
Observations	837			837		
Marginal R^2^/ Conditional R^2^	0.137/0.672			0.089/0.669		

Gender: 1/2 = female, −1/2 = male; PD = psychological distress; LOC = locus of control; wp.DNES = within-person daily emotional state; bp.DNES = between-person daily emotional state; DSUD = daily smartphone use duration; DSUF = daily smartphone use frequency.

**Table 3 behavsci-15-01121-t003:** Multilevel linear model (Model 2) results for daily smartphone use duration and frequency.

	DSUD			DSUF		
Predictors	*β*	*t*	*p*	*β*	*t*	*p*
PD	0.09	0.75	0.451	0.01	0.20	0.845
LOC	0.17	1.40	0.161	0.22	2.12	0.034
wp.DNES	0.03	1.55	0.122	0.01	0.37	0.711
bp.DNES	−0.04	−0.28	0.783	−0.04	−0.31	0.759
Age	0.34	2.33	0.020	0.17	1.26	0.209
Gender	0.04	0.29	0.772	0.15	1.26	0.209
PD:LOC	0.27	1.79	0.074	0.47	3.44	0.001
wp.DNES:LOC	−0.02	−1.01	0.312	−0.03	−1.30	0.194
	Random Effects					
σ^2^	10,678.22			4191.17		
τ_00_	16,215.99			5392.40		
ICC	0.60			0.56		
N	37			37		
Observations	837			837		
Marginal R^2^/ Conditional R^2^	0.168/0.670			0.216/0.657		

Gender: 1/2 = female, −1/2 = male; PD = psychological distress; LOC = locus of control; wp.DNES = within-person daily emotional state; bp.DNES = between-person daily emotional state; DSUD = daily smartphone use duration; DSUF = daily smartphone use frequency.

## Data Availability

The raw data supporting the conclusions of this article will be made available by the authors on request.

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
