# Peer review of "Differential Effects of Short- and Long-Term Negative Affect on Smartphone Usage: The Moderating Role of Locus of Control"

_behavsci, 2025, doi:10.3390/bs15081121_

Round 1

Reviewer 1 Report

Comments and Suggestions for Authors

The manuscript, “Differential Effects of Short- and Long-Term Negative Affect on Smartphone Usage: The Moderating Role of Locus of Control,” addresses a timely and relevant topic at the intersection of affective psychology and digital behavior. The authors aim to advance Compensatory Internet Use Theory (CIUT) by integrating both state- and trait-level negative affect and exploring the moderating role of locus of control (LOC). While the study makes a valuable empirical contribution by combining longitudinal diary data with objective smartphone tracking, several limitations in the theoretical framework, measurement strategy, and interpretation of results must be addressed before the manuscript can be considered for publication.

The introduction and theoretical framework are grounded in relevant prior literature, particularly in relation to CIUT. However, the conceptual rationale would benefit from a broader and more critical engagement with adjacent theoretical models. Specifically, the manuscript currently lacks a discussion of alternative frameworks such as Uses and Gratifications Theory or dual-system models of behavior, which could enhance the theoretical richness and explanatory power of the proposed model. Moreover, although the authors briefly differentiate between smartphone use duration (DSUD) and frequency (DSUF), the theoretical justification for this distinction is underdeveloped. Given that later analyses yield divergent patterns for DSUD and DSUF, the authors should more explicitly ground this distinction in prior research on habit strength, impulsivity, and digital self-regulation.

An additional shortcoming in the literature review is the limited engagement with relevant recent studies that would substantiate the psychological mechanisms proposed. For instance, the study by Wickord and Quaiser-Pohl (2025), which examines the roles of flow experience and media multitasking in different types of smartphone use and their links to problematic usage patterns, offers a useful empirical backdrop for interpreting habitual checking and duration-based usage. Similarly, the work of Yıldız Durak (2018), which links locus of control, social media use, and nomophobia in adolescents, would offer valuable empirical support for the conceptual role of LOC as a predictor of maladaptive smartphone behaviors. These references should be integrated into the theoretical and discussion sections to strengthen both the conceptual framing and the practical relevance of the findings.

The empirical design is methodologically sound, with an appropriate multilevel modelling approach to account for nested data and separate within- and between-person effects. The sample size (N = 37) and one-month diary duration are reasonable for this type of ecological measurement study. The use of objectively logged smartphone behavior is a significant strength, mitigating biases associated with self-report. However, a major limitation lies in the assessment of daily negative emotional state (DNES), which is operationalized using a single reversed-coded item. While the authors reference participant burden as a rationale, this approach substantially compromises construct validity and reduces interpretability of the null findings for within-person emotional effects. At minimum, this limitation must be more thoroughly acknowledged in the discussion, and the authors should consider using validated short-form scales in future work (e.g., PANAS short form).

The results are clearly reported and statistically robust. Tables and figures are appropriately used, and the interaction effect between PD and LOC on smartphone use frequency is an important and novel finding. The result that psychological distress increases usage frequency in individuals with external LOC and decreases it in those with internal LOC is well-visualized and conceptually coherent. However, the discussion does not sufficiently consider competing explanations, such as avoidance-based coping strategies, social reassurance seeking, or digital procrastination, which have been associated with both negative affect and external locus of control. Without this, the interpretation risks overstating the uniqueness of the proposed mechanism.

The quality of academic English throughout the manuscript is acceptable but would benefit from revision. Several sentences are unnecessarily complex or passive, particularly in the introduction and discussion sections. A thorough copyediting pass is recommended to improve clarity and academic tone.

In conclusion, the manuscript offers a potentially valuable contribution to research on emotional regulation and smartphone use, especially through its methodological rigor and integration of objective behavioral data. I recommend a Major Revision.

References:

Yıldız Durak, H. (2018). What would you do without your smartphone? Adolescents’ social media usage, locus of control, and loneliness as a predictor of nomophobia.

Wickord, L. C., & Quaiser-Pohl, C. (2025). The Role of Flow and Media Multitasking for Problematic Smartphone Use and the different Types of Smartphone Use. Computers in Human Behavior, 108583.

Reviewer 2 Report

Comments and Suggestions for Authors

Thank you for the opportunity to review this manuscript, which examines the relationship between negative affect and smartphone use. By considering both short-term emotional states and long-term affect over an one-month period, as well as individual psychological resources (locus of control), the study offers a valuable contribution to the existing literature in this area.

However, there are several issues that need to be addressed.

Introduction:

The introduction is well-structured. However, in several instances, the authors use broad phrases such as “many studies” and “prior research” without providing sufficient references to support the corresponding statements (or cite only a single source). It is recommended that these statements be supported with additional, relevant literature.

Results:

The correlation results should be described in greater detail, particularly concerning the associations between smartphone usage, on the one hand, and negative affect (as measured by DNES and PD) and age, on the other.

Additionally, it is important to clarify whether demographic variables, such as age and gender, were accounted for or controlled in the analyses, and to consider the extent to which these factors may have influenced the obtained results.

Discussion

The discussion could be expanded with a more detailed reflection on possible explanations for the lack of association between PD and smartphone use.

Additionally, it is necessary to ensure consistency between the hypotheses outlined at the end of the Introduction and the way they are addressed in the Discussion. For instance, the paragraph addressing the absence of between-person association between PD and smartphone use refers to Hypothesis 2, not Hypothesis 3. A similar clarification is needed in the paragraph discussing the moderating role of locus of control.

Round 2

Reviewer 1 Report

Comments and Suggestions for Authors

Thank you very much for the revision. The article now has a clear structure and coherence. The reference list is still incomplete.
